# Carboxymethyl Cellulose Entrapped in a Poly(vinyl) Alcohol Network: Plant-Based Scaffolds for Cartilage Tissue Engineering

**DOI:** 10.3390/molecules26030578

**Published:** 2021-01-22

**Authors:** Jirapat Namkaew, Panitporn Laowpanitchakorn, Nuttapong Sawaddee, Sirinee Jirajessada, Sittisak Honsawek, Supansa Yodmuang

**Affiliations:** 1Excellence Center for Advanced Therapy Medicinal Products, King Chulalongkorn Memorial Hospital, Pathumwan, Bangkok 10330, Thailand; namkaew.j@gmail.com (J.N.); nuttapong_tontoey@hotmail.com (N.S.); 2Biomaterial Engineering for Medical and Health Research Unit, Chulalongkorn University, Pathumwan, Bangkok 10330, Thailand; antpnp@gmail.com; 3Biology Program, Faculty of Science, Buriram Rajabhat University, Muang, Buriram 31000, Thailand; sirinee.ym@bru.ac.th; 4Osteoarthritis and Musculoskeleton Research Unit, Faculty of Medicine, Chulalongkorn University, Pathumwan, Bangkok 10330, Thailand; sittisak.h@chula.ac.th; 5Research Affairs, Faculty of Medicine, Chulalongkorn University, Pathumwan, Bangkok 10330, Thailand

**Keywords:** poly(vinyl) alcohol, sodium carboxymethyl cellulose, scaffolds, tissue engineering

## Abstract

Cartilage has a limited inherent healing capacity after injury, due to a lack of direct blood supply and low cell density. Tissue engineering in conjunction with biomaterials holds promise for generating cartilage substitutes that withstand stress in joints. A major challenge of tissue substitution is creating a functional framework to support cartilage tissue formation. Polyvinyl alcohol (PVA) was crosslinked with glutaraldehyde (GA), by varying the mole ratios of GA/PVA in the presence of different amounts of plant-derived carboxymethyl cellulose (CMC). Porous scaffolds were created by the freeze-drying technique. The goal of this study was to investigate how CMC incorporation and crosslinking density might affect scaffold pore formation, swelling behaviors, mechanical properties, and potential use for engineered cartilage. The peak at 1599 cm^−1^ of the C=O group in ATR–FTIR indicates the incorporation of CMC into the scaffold. The glass transition temperature (T_g_) and Young’s modulus were lower in the PVA/CMC scaffold, as compared to the PVA control scaffold. The addition of CMC modulates the pore architecture and increases the swelling ratio of scaffolds. The toxicity of the scaffolds and cell attachment were tested. The results suggest that PVA/CMC scaffolding material can be tailored in terms of its physical and swelling properties to potentially support cartilage formation.

## 1. Introduction

Articular cartilage is a specialized connective tissue covering the ends of long bones. It has unique viscoelastic properties and exhibits time-dependency in its stress–strain response. It serves as a shock absorber as it is tough but highly deformable and lubricious for low friction. After injury, the dense matrix and poor vascularization of cartilage contribute to the lack of intrinsic repairability by preventing mature chondrocytes or progenitor cells from bone marrow migration to defect areas [1]. Without proper treatment, cartilage defects may progress to osteoarthritis (OA) and eventually require total joint replacement. Although better overall health care is contributing to a steady rise in the number of elderly people worldwide, OA remains a challenge for the elderly, who face limited mobility. Better OA treatment will contribute to improving the quality of life of this elderly population including those who suffer from trauma, sports injuries, and repetitive use of joints [2].

The available treatment options for chondral defects include pain medications and arthroscopic surgery, e.g., abrasion, debridement, and microfracture [2]. Despite these promising procedures being minimally invasive, they rarely restore structure and function of injured cartilage to the level of native tissue. Cell-based cartilage tissue repair aims at replacing and regenerating damaged cartilage without complications associated with donor-site morbidity from autografts and the risk of disease transmission or induction of immune response from allografts [3]. Over the past 20 years, researchers have been in search of biomedical polymers in order to construct the template for cell proliferation and tissue development.

Synthetic biomaterials such as polycaprolactone [4], polyvinyl alcohol [5], and polylactic acid [6] have well-defined mechanical and rheological properties and transport of molecules by modulating degree of substitution and crosslinking density [7]. However, these materials are inert and lack biological cues that can promote cell adhesion, proliferation, and chondrogenic differentiation. Natural biomaterials such as collagen [8], silk [9], Matrigel^®^(Corning, NY, USA) [10], chitosan [11], and hyaluronic acid [12] containing biological cues similar to the extracellular matrix have been used to make scaffolds. Although natural materials are biocompatible and biodegradable, these materials generally have poor mechanical properties and are difficult to handle. Recently, the utilization of renewable resources and eco-friendly materials, especially from plants, has gained increasing attention in medical innovation (cosmetic, foods, drug capsules) [13,14].

The natural polymer of choice in this study is carboxymethyl cellulose (CMC). CMC is a polysaccharide polymer containing hydroxyl groups, which are replaced by sodium carboxymethyl groups in C2, C3, and C6 of glucopyranose [15]. CMC has been widely used as a dispersion agent in the paint industry, a thickener in foods, and a stabilizer in pharmaceutical and medical products [16]. Highly negatively charge from -COO- and -O- groups in aqueous solution enables CMC to attract water and create a hydrated environment similar to the extracellular matrix. In order to leverage the use of CMC in cartilage tissue engineering applications, scaffolds made from CMC have to withstand loads in joints after implantation. Combining CMC with more stable and controllable synthetic biomaterials will address the mechanical limitation of the naturally derived polymer and preserve biological properties of CMC in scaffold constructs. Polyvinyl alcohol (PVA) is a hydrophilic synthetic polymer that can be a non-biodegradable polymer after chemical crosslinking. It is biocompatible and has FDA approval for clinical uses, including wound dressing, drug delivery vehicles, and tissue engineering application [17,18].

The objective of this study was to investigate how CMC incorporation and crosslinking density might affect PVA/CMC scaffold formation. We hypothesized that incorporation of CMC increases hydrated environment in scaffolds to promote cell attachment without cytotoxicity. To test this hypothesis, PVA was crosslinked with glutaraldehyde (GA) by varying mole ratios of GA/PVA in the presence of different PVA/CMC mass ratios. Pore architecture, swelling behaviors, and mechanical properties of PVA/CMC porous scaffolds were investigated. The composition and thermal properties of composite scaffolds were characterized by ATR-Fourier transform infrared spectroscopy (ATR-FTIR) and differential scanning calorimetry (DSC), respectively. Chondrocytes were seeded into the scaffold to evaluate cytotoxicity for 7 days. Through this work, we provide valuable insights into the role of synthetically and naturally derived polymers in controlling properties of porous scaffolds. Crosslinking density in conjunction with CMC entrapment can be used to tailor mechanical properties and structure of the scaffold for engineered cartilage.

## 2. Results

### 2.1. Appearances of Hydrogels after Fabrication

PVA/CMC scaffolds were fabricated by combining PVA and CMC at different mass ratios (PVA/CMC). The degree of crosslinking was varied and indicated by the mole ratio of GA and PVA (GA/PVA). The PVA control group, P1C0, was able to form hydrogels in all crosslinking ratios, while P5C1 and P3C1 groups failed to form hydrogel at crosslinking ratio of 0.05 (Figure 1).

### 2.2. Fourier Transform Infrared Spectroscopy (FTIR)

The scaffolds P1C0, P5C1, and P3C1 at a GA/PVA crosslinking ratio of 0.2 were chosen for ATR-FTIR analysis. FTIR spectra of PVA powder showed the stretching O-H of the hydroxyl group at 3360–3321 cm^−1^. The C=O of acetyl group (1715–1733 cm^−1^) was observed. These peaks are specific for the remaining acetate groups of polyvinyl acetate, which is a precursor for polyvinyl alcohol production (Figure 2). Crystallinity of PVA before dissolving was clearly observed only in PVA powder at 1140 cm^−1^. A shift in peak at 1570 cm^−1^ was observed in P1C0, P5C1, and P3C1 scaffolds, which crosslinked with glutaraldehyde. The peaks at 2851 and 2871 cm^−1^ indicated that the C-H resulted from glutaraldehyde crosslinking in all scaffolds. The major characteristic peak of CMC is the stretching C=O of the COONa group at 1592 cm^−1^**,** which was shifted to 1599 cm^−1^, as demonstrated in P5C1 and P3C1 groups.

### 2.3. Differential Scanning Calorimetry (DSC)

The thermal properties of scaffolds could be modulated by varying PVA and CMC contents and crosslinking density. The onset of the glass transition temperature (T_g_) ranged from 70 to 85 °C, as demonstrated by a sudden decrease of the heat flow in the heating curve (Figure 3A). In the P1C0 control group, the GA/PVA crosslinking ratio of 0.4 exhibited higher T_g_ (84.2 °C) compared to other crosslinking ratios (78.1 °C for GA/PVA = 0.2 and 72.0 °C for GA/PVA = 0.1). When CMC was incorporated into scaffolds, a slight decrease in T_g_ was detected in P5C1 and P3C1 groups.

The melting temperature (T_m_) was located at another decrease of the heat flow around 150 °C (Figure 3A). When the crosslinking ratio decreased from 0.2 to 0.1, T_m_ increased to 156.5 °C for P1C0, 154.6 °C for P5C1 and 157.7 °C for P3C1 (Appendix A). At a crosslink ratio of 0.4, the T_m_ values of all groups were unable to be determined, as indicated by dashed lines in Figure 3A. When the crosslinking ratio decreased to 0.1, the crystallization temperature (T_c_) of the P1C0 control group increased, as demonstrated by the highest T_c_ of 142.4 °C (Figure 3B and Appendix A). The addition of CMC tended to increase T_c_ of P5C1 and P3C1 groups, but varying crosslinking ratios did not affect T_c_ of these two groups. Specifically, in the P3C1 group, crosslinking ratios of 0.2 and 0.4 showed similar T_c_ (135.9 °C for crosslinking ratio 0.2 vs. 136.6 °C for 0.4). Surprisingly, T_c_ of P5C1 group at a crosslinking ratio of 0.4 was not detected as indicated by dashed line in Figure 3B.

### 2.4. Micromorphological Assessment

The porosity of PVA/CMC scaffolds was visualized by scanning electron microscopy (SEM). P1C0 control scaffolds at a GA/PVA crosslinking ratio of 0.4 displayed thick walls and small pore sizes (Figure 4). When the crosslinking ratio decreased to 0.2 and 0.1, thinner walls were observed. A porous structure of the P1C0 control group was not observed for the lowest crosslinking ratio, 0.05.

Interestingly, incorporation of CMC resulted in larger pore sizes in P5C1 and P3C1 groups. The P3C1 groups at a crosslinking ratio of 0.4 exhibited pore sizes of 50–80 μm, which were larger than those observed in the P5C1 group and P1C0 control group. The pore sizes of the P5C1 group decreased when the crosslinking ratio was reduced to 0.2 and 0.1. In contrast, the P3C1 group exhibited larger pore sizes (50 μm) and thin walls at a crosslinking ratio of 0.1 (Figure 4).

### 2.5. Mechanical Analysis of Scaffolds

The compressive modulus of the scaffolds was evaluated (Figure 5). The highest Young’s modulus (344.74 ± 52 kPa) was observed in the P1C0 control group at a GA/PVA crosslinking ratio of 0.4 (Figure 5 and Appendix A). CMC incorporation resulted in P5C1 and P3C1 groups having a significantly lower Young’s modulus compared to the control group. At a crosslinking ratio of 0.1, when CMC content increased, Young’s modulus decreased from 175.40 ± 24.3 kPa in the control group to 23.73 ± 5.93 and 14.77 ± 1.49 kPa in the P5C1 and P3C1 groups, respectively. The decrease in Young’s modulus was also observed for the crosslinking ratio of 0.2.

Interestingly, the P5C1 group at a crosslinking ratio of 0.2 and P3C1 group at a crosslinking ratio of 0.4 displayed a similar Young’s modulus (~100 kPa). Based on the similarity in Young’s modulus for these two groups, they were chosen for cell seeding and cell viability testing.

### 2.6. Swelling Behaviors of PVA/CMC Scaffolds

The swelling ratio and swelling rate of PVA/CMC scaffolds were evaluated in PBS. P1C0 control scaffolds did not show significant differences in swelling ratio when GA/PVA crosslinking ratios were decreased from 0.4 to 0.05 (Figure 6A). Effects of incorporation of CMC on water absorption into scaffolds was demonstrated by a significant change in swelling ratio in the P5C1 and P3C1 groups, especially at the GA/PVA crosslinking ratio of 0.1 (Figure 6B,C). At the lowest crosslinking ratio, 0.05, only the P1C0 control group could form hydrogel (Figure 1), and swelling behaviors of the scaffolds could be observed (Figure 6A, ▽).

After 24 h (Figure 6D), the highest swelling ratio was observed in P3C1 at the crosslinking ratio of 0.1 (12.76 ± 0.82). An increase in GA/PVA crosslinking ratio to 0.4 decreased water absorption in both CMC incorporated groups, P5C1 and P3C1. However, swelling behaviors of control PVA did not change regardless of crosslinking ratio (Figure 6D). In the P5C1 and P3C1 groups, the GA/PVA crosslinking ratio of 0.1 showed a significant increase in swelling behaviors compared to GA/PVA crosslinking ratios of 0.2 and 0.4. The highest swelling rate was observed in the P3C1 group at a crosslinking ratio of 0.1 (20.84 ± 2.35 h^−1^) followed by the swelling rate of the P5C1 group at a crosslinking ratio of 0.1 (15.91 ± 0.86 h^−1^) (Figure 6E). The swelling rate corresponded with the swelling ratio in the P3C1 group, which contained a higher water content and absorbed water into scaffolds faster than other groups. In addition, PVA/CMC scaffolds were submerged in PBS to investigate the in vitro degradation for 8 weeks, with results reported as percentage of remaining weight. Significant mass loss could not be detected in all groups (Appendix A).

### 2.7. Assessment of Cell Distribution and Cytotoxicity of PVA/CMC Scaffolds

A LIVE/DEAD cell viability assay was performed on day 7 post-seeding. Chondrocytes showed a homogeneous distribution in P5C1 scaffolds at a crosslinking ratio of 0.2 and P3C1 scaffolds at a crosslinking ratio of 0.4. Green live cells and red auto-fluorescence of scaffolds could be observed in both groups (Figure 7A).

Viability percentages of chondrocytes cultured in PVA/CMC scaffold extracts were higher than 70%. The viability percentage of P5C1 at a crosslinking ratio of 0.2 was 92.32%, and that of P3C1 at a crosslinking ratio of 0.4 was 86.91% (Figure 7B), indicating non-toxicity of PVA/CMC scaffolds against chondrocytes. Growth medium and 10% DMSO served as negative and positive controls for cytotoxicity, respectively.

## 3. Discussion

Cell-based cartilage tissue repair in conjunction with biopolymers provides treatment options by creating functional tissues to replace diseased and damaged ones. Our study developed porous scaffolds for potential use in cartilage tissue engineering by entrapping plant-based carboxymethylcellulose (CMC) in a polyvinyl alcohol (PVA) network resulting in PVA/CMC scaffolds. We hypothesized that incorporation of CMC increases hydrated environment in scaffolds to promote cell adhesion without cytotoxicity. CMC is a carbohydrate-derived polymer with excellent water absorption and biodegradable properties, while PVA is an FDA-approved synthetic polymer. We demonstrated how to control mechanical properties, swelling behaviors, and potential use of scaffolds as cell carrier for cartilage tissue development via incorporation of CMC into PVA and varying degrees of glutaraldehyde crosslinking.

The advantages of using glutaraldehyde crosslinking are that thermal treatment and catalysts are not required in this study, making scaffold production easily to handle. However, the absence of a catalyst or thermal treatment during the crosslinking reaction resulted in slow hydrogel formation or failure to form hydrogel [19]. In this study, PVA/CMC solution took 10–14 days at 25 °C to become hydrogel. Adding hydrochloric acid (HCl) as a catalyst allows faster crosslinking time, and this can be achieved by directly mixing PVA solution with HCl [20,21] and exposing PVA nanofibers and film to glutaraldehyde-hydrochloric acid vapor [22,23]. The crosslinking used in this study was based on mole ratio, which takes functional groups of GA and PVA into account. Swelling ratio decreased for all scaffolds at high GA/PVA crosslinking ratios, similarly to results reported for PVA films and hydrogels [20,24,25]. At a high GA/PVA crosslink ratio, glutaraldehyde used up hydroxyl groups on PVA molecules. A decrease in hydroxyl groups led to an increase in hydrophobicity of scaffolds, which resulted in less water absorption, as demonstrated by the very low swelling behaviors of all scaffolds at GA/PVA crosslinking ratio of 0.4. Besides GA, other crosslinking agents, such as epichlorohydrin [26] and citric acid [27], were used to crosslink PVA and CMC mixture under alkaline or acidic conditions at high temperature. These crosslinking reactions resulted in different products: crosslinked PVA–PVA, PVA–CMC, and CMC–CMC.

Swelling behaviors of PVA/CMC scaffolds could be explained by two counteracting forces, which are swelling pressures from CMC and tensile strengths from PVA networks. PVA networks were formed via acetal bridges that linked hydroxyl groups on PVA with aldehyde groups on glutaraldehyde, while CMC was trapped inside the networks absorbing water. ATR-FTIR analysis of PVA/CMC scaffolds showed a decrease in transmittance intensity of the stretching O-H (at 3321–3360 cm^−1^) after glutaraldehyde crosslinking. A decrease in the intensity might occur as a result of the formation of acetal bridges, while the stretching O-H bands in control PVA powder was maintained at a high intensity. The clear signal at 1599 cm^−1^ of P3C1 and P5C1 confirmed incorporation of CMC in the PVA hydrogel network.

P3C1 scaffolds with high CMC content showed a significantly higher swelling ratio compared with P5C1 and control groups, indicating a highly hydrophilic nature of CMC. Interestingly, P5C1 and P3C1 groups contained the same amount of PVA as the P1C0 control group (12.5% (*w*/*v*)), but they were dissolved and could not form hydrogel at a GA/PVA crosslinking ratio of 0.05. It is possible that P5C1 and P3C1 groups did not contain sufficient crosslinking networks, covalent bonds generated from carbonyl groups (-CHO) of glutaraldehyde and hydroxyl groups (-OH) of PVA. Specifically, acetal bridges in P5C1 and P3C1 hydrogels could not withstand an increase in swelling pressure generated from negatively charged (carboxyl groups, COO^−^) of CMC, resulting in hydrogel falling apart. An increase in GA/PVA crosslinking ratio up to 0.1 was sufficient to counteract swelling pressure in P5C1 and P3C1 groups.

This study demonstrated the use of freeze-drying to create the pore structure of three-dimensional scaffolds. As seen in SEM images (Figure 4), incorporating CMC into PVA hydrogel increased the pore size of scaffolds. When hydrogel was soaked in water before the freeze-drying step, both P5C1 and P3C1 hydrogels absorbed water more than the PVA control group, resulting in more ice crystals inside scaffolds. After sublimating, larger pore cavities were left behind in P5C1 and P3C1 scaffolds compared to PVA control scaffolds.

In addition, larger pore sizes were clearly observed at the high GA/PVA crosslinking ratio of 0.4, while at a low crosslinking ratio porous structure was not clearly present in scaffolds. One plausible explanation for this observation is related to low crosslink density of the hydrogel network, resulting in collapse of pore walls. However, it is interesting to note that pore size of the P3C1 group increased at the low crosslinking ratio of 0.1. With freeze-drying, PVA/CMC scaffolds exhibited 12- to 15-fold lower swelling ratios after reach equilibrium compared to the scaffolds that did not proceed to the freeze-drying process, without the pore formation step (Appendix A). The low swelling ratio observed in freeze-dried scaffolds in this study was affected by the drying process, which possibly interfered with hydrophilic groups on polymers to attract water. Nevertheless, newly formed large pores and good connectivity led to an ideal scaffold for cell seeding generated from freeze-drying, which contributed to scaffold swelling through water-filled pores. The porous structure in a scaffold is important for cartilage tissue engineering to quickly absorb joint liquid and to allow cell migration and ingrowth into scaffolds [8]. In our study, the P5C1 scaffold at a crosslinking ratio of 0.2 and P3C1 scaffold at a crosslinking ratio of 0.4 exhibiting large pore sizes of around 50–100 μm were chosen to study cell viability.

Reduction of the crosslinking ratio from 0.4 to 0.2 did not affect mechanical properties of P1C0 control groups. It is possible that the acetal bridge easily formed in P1C0 hydrogel because CMC was not present in the hydrogel mixture to hinder the crosslinking process. Therefore, a crosslink ratio of 0.2 for the P1C0 group was sufficient to increase mechanical properties of the scaffolds. Interestingly, incorporation of CMC into PVA at a mass ratio PVA/CMC = 5:1 significantly modulated mechanical properties of porous scaffolds as seen in the increment of Young’s modulus when crosslinking ratio was increased (Figure 5). While CMC content increased to one-third of PVA mass, mechanical properties of P3C1 scaffolds did not change regardless of increasing GA/PVA ratio (Figure 5). It is possible that glutaraldehyde inefficiently reacted with hydroxyl groups on PVA due to high CMC contents in P3C1 hydrogel mixture preventing acetal bridge formation. This study suggested that modulation of mechanical properties by glutaraldehyde crosslink occurred when CMC content at one-fifth of PVA mass was incorporated into the hydrogel mixture.

Ideally, scaffold for cartilage repair should mechanically resist load in in vivo joints and provide a three-dimensional structure for cell growth and nutrient transport and hydrated environment similar to native tissue with water content in a range of 70% to 80% [28,29]. In our study, two formulas of PVA/CMC scaffolds were chosen to study chondrocyte viability. The first formula was for P5C1 scaffolds at a GA/PVA crosslinking ratio of 0.2, and the second was for P3C1 scaffolds at a GA/PVA crosslinking ratio of 0.4. These two scaffolds exhibited similar water content to native cartilage of approximately 70% (Appendix A) and similar mechanical properties.

Previously, we seeded chondrocytes on P1C0 control scaffolds and found that cells could not adhere to the scaffold and sedimented at the bottom of the cell culture tubes. One plausible explanation for poor cell attachment of the control scaffolds is their hydrophobic surface that results from a decrease of hydroxyl groups during glutaraldehyde crosslinking. Kim et.al. demonstrated that biomolecules in serum containing media could modulate cell binding affinity and shape [30]. In the current study, cell-scaffold interaction was minimally influenced by serum protein absorption because serum-free growth medium was used during 7-day post-cell seeding. The variety of substrates was investigated the effects of charges on cell adhesion. It has been generally accepted that positively charged substrates bind negatively charged cell membranes via electrostatic interaction [31]. In addition, Webb et al. demonstrated greater cell attachment on hydrophilic and positively charged amine-modified surfaces compared to hydrophobic surfaces [32]. We have limited information on characterization of PVA/CMC scaffolds in terms of the mechanism behind cell adhesion. At physiological pH 7.4, PVA/CMC scaffolds are expected to contain negatively charged carbonyl groups (-CH_2_COO-) from CMC and non-charged hydroxyl groups (-OH) on PVA, which contribute to hydrophilicity of the scaffold. Further study will be necessary to elucidate the underlining mechanism of cell adhesion to the PVA/CMC scaffold, which is one of criteria of biocompatibility besides normal cell function, migration to scaffolds, and proliferation before laying down a new matrix [33].

PVA was not chemically covalent with CMC, but these two polymers co-existed via interpenetrating network (IPN) to form hydrogel without any covalent bonds between them [34]. Our study demonstrated that hydroxyl groups on PVA formed covalent bonds with glutaraldehyde, while CMC entangled the polymer network. No differences were found in solid-state NMR spectra between P3C1 and P1C0 control groups (Appendix A). New functional groups were also not present in the NMR spectra of the P3C1 group. Unlike IPN hydrogel, single network hydrogels have a slow response to water absorption [35]. We demonstrated that the P1C0 control group exhibited an 8-times slower swelling rate compared to the P3C1 group at a crosslinking ratio of 0.1. Fast swelling response promotes cell seeding efficiency, due to quick cell absorption into scaffolds. It has been previously reported that IPN was used to construct cartilage scaffolds using alginate and chitosan (natural polymers) [36] as well as gelatin and polycaprolactone-polyethylene glycol (natural and synthetic polymers) [37]. The PVA/CMC scaffolds in this study displayed structural advantages through mechanical stability of the PVA network and high crosslinking density, and at the same time they maintained a hydrated environment through incorporating CMC.

PVA control scaffolds might have a rigid amorphous structure, which requires higher temperature to heat up its polymer chains from brittle-solid to viscous rubbery state [38,39]. CMC incorporation and low crosslinking ratio tended to decrease T_g_ of the PVA/CMC scaffolds, as demonstrated by low T_g_ in the P5C1 and P3C1 groups. It is possible that CMC caused PVA polymer chains to move around easily. Previous studies reported polymer blend decreased T_g_ [40]. In those studies, an increase in chitosan contents in PVA solution slightly decreased T_g_ of the thin film blend. High density of crosslinking, the acetal bridge network in this case, reduced the freedom of motion of the segments of the polymer chains and thus increased T_g_ [38]. We demonstrated that incorporation of CMC possibly impeded acetal bridge formation. In addition, our study showed that T_g_ could be restored by increasing GA/PVA crosslinking ratio.

At a GA/PVA crosslinking ratio of 0.4, all groups did not show T_m_ in DSC curves (Figure 3A). The absence of T_m_ peaks in all the three groups at high crosslinking ratio might be from their amorphous structure, which prevented the detection of melting points [41]. Interestingly, only P5C1 scaffolds at a GA/PVA crosslinking ratio of 0.4 did not show both T_m_ and T_c_. However, we cannot explain why scaffolds at low crosslinking ratios of 0.2 and 0.1 exhibited higher T_m_ than scaffolds at the high crosslinking ratio of 0.4 (Appendix A). Generally, polymer chains are less restricted at low crosslinking density, which allows them easily to slide across one another. In addition, we found that addition of CMC tended to increase T_m_ of the porous scaffolds. The higher melting temperature of the scaffold was possibly influenced by CMC (274 °C), which has a higher melting point than PVA (~200 °C).

Crystallization is an exothermic process that occurs when polymers move from a disorder state after melting to a more ordered crystalline state and release the amount of energy difference between those two states while cooling down [39]. Crosslinking restricts chain motion that could prevent crystallization after cooling down. In the P1C0 control group, GA/PVA crosslinking ratio increased from 0.1 to 0.4, resulting in a decrease of T_c_. A similar observation of low T_c_ and T_m_ after an increase in crosslinking ratio was reported in poly(l-lactide) (PLLA) electrospun nanomat [42] and poly(butylene succinate) hydrogel [43]. Our study showed that T_c_ of PVA/CMC scaffolds was independent from crosslinking density, as demonstrated in similar T_c_ of P5C1 and P3C1 groups ranging from 133 to 136 °C regardless of crosslinking ratio.

Functional cartilage tissue requires incorporation of living cells into scaffolds, which serve as structural support for cell adhesion and new matrix deposition. The PVA/CMC scaffolds derived from synthetic and natural polymers in this study not only provided porous structure for chondrocyte seeding but also acknowledged green chemistry movement. Compared to other fields, tissue engineering does not represent green and eco-friendly technologies because traditional scaffold fabrication processes still employ toxic solvents [44]. The use of plant-derived polysaccharides blended with PVA has emerged in biomedical applications, such as wound healing hydrogel [45], drug release hydrogel film [26,27], and cryogel scaffolds for bone tissue engineering [46]. Biocompatibility, biodegradability, and plant proteins abundant in nature (soy, zein, gluten) attract much attention as alternative scaffolding biomaterials for regenerative medicine. Green chemistry-inspired scaffolds in our study reduced the use of toxic and harmful materials and method. Cellulose-derived biomaterials were incorporated to create hydrated microenvironment of scaffolds without acidic conditions in the crosslinking process. Although glutaraldehyde was used as a crosslinking agent, its toxicity was inactivated by glycine during the preparation of scaffolds.

In conclusion, porous scaffolds were developed from CMC entrapped in a PVA network. The data presented in this work suggest that the scaffolds can be tailored in terms of pore sizes, mechanical properties, and water contents by modulating crosslinking density and CMC contents. We demonstrated that PVA/CMC scaffolds possessed biological cues from plant-derived polymers and structural supports from synthetic polymers. The scaffolds showed biocompatibility with chondrocytes and may have great potential for applications in cartilage regeneration.

## 4. Materials and Methods

### 4.1. Chemicals and Reagents

Polymers used for scaffold fabrication were polyvinyl alcohol (M_W_ = 85,000–124,000 g/mol) and sodium carboxymethyl cellulose (MW = ~90,000 g/mol, degree of substitution 0.7). These polymers were purchased from Sigma-Aldrich (St. Louis, MO, USA). Glutaraldehyde (03965) was purchased from Loba Chemie (Mumbai, India). High glucose DMEM (D777) used in tissue culture was purchased from Sigma-Aldrich (St. Louis, MO, USA). Gibco^™^ HEPES (Waltham, MA, USA) (15630-080) and Gibco^™^ antibiotic-antimycotic (Waltham, MA, USA) (A5955) were purchased from ThermoFisher Scientific (Waltham, MA, USA). Serum replacement solution (SR-100) was purchased from PeproTech (Rocky Hill, NJ, USA). TrypLE^TM^ Express Enzyme (Waltham, MA, USA) (12604021) and Invitrogen^™^ LIVE/DEAD^®^ viability/cytotoxicity kit (Waltham, MA, USA) (L3224) were purchased from ThermoFisher Scientific (MA, USA).

### 4.2. Preparation of Porous PVA/CMC Scaffolds

Polyvinyl alcohol (PVA) was mixed with carboxymethyl cellulose (CMC) at different mass ratios (Table 1). CMC (1.25 g or 2.09 g) was dissolved in deionized (DI) water, mixed well, and heated at 80 °C overnight. PVA (6.25 g) was slowly added into CMC thick solution and heated at 80 °C overnight to increase homogeneity. The polymer mixture was centrifuged at 1500 rpm for 2 min. Glutaraldehyde (GA) was added to the polymer mixture (Appendix A). PVA polymer solutions without CMC served as control groups. Summaries of the PVA/CMC mass ratios and GA/PVA mole ratios are shown in Table 1.

Five milliliters of homogeneous mixture were poured into 6-well plates. The plates were sealed with Parafilm and left in a chemical hood for 14 days. Hydrogels were soaked in 5 L of DI water for 48 h. The hydrogels in 6-well plates were transferred to −20 °C and kept overnight. Cylinder constructs were taken from the frozen hydrogel using a biopsy punch 8 mm in diameter, cut to a desired height of 5 mm, and washed with DI water for 2 days with 3 changes of DI water per day. To inactivate uncrosslinked glutaraldehyde, hydrogel constructs were incubated in glycine (50 mM) on a rocker overnight and washed with distilled water for 24 h. The constructs were frozen at −80 °C for 18 h with a cooling rate of 1 °C/minute and transferred into a freeze-dryer (Alpha-4, Martin Christ, Germany) for 24 h. Dried scaffolds were soaked in absolute ethanol, air-dried, and exposed to UV light for 15 min. The samples were stored in sterile bottle until use in cell seeding experiment.

### 4.3. Characterization of Porous Scaffold

For FTIR measurement, scaffolds were cut into 2 × 2 mm² samples and analysed in ATR mode. FTIR spectra were obtained in the range of wavenumber from 4000 to 400 cm^−1^. The spectrum was averaged over 64 scans with 4.0 cm^−1^ resolutions (Perkin Elmer, Spectrum One, USA). For DSC characterization, scaffolds (5–10 mg) were heated from 25 to 220 °C under N_2_ at a linear heating rate of 10 K/min (DSC 204 **F1** Phoenix^®^, Selb, Germany). For solid state NMR analysis, the 1D ^13^C Cross-Polarization Magic Angle Spinning (CP/MAS) spectra of scaffolds were acquired using a 400MHz solid-state Nuclear Magnetic Resonance spectrometer AVANCE III, Bruker (Billerica, MA, USA) at 100 MHz, with 4mm PH MAS 400WB BL4 N-P/H DVT probe. The 2341 scans were induced and collected over a spectral width of 30 kHz, with a recycle delay of 3 s. Glycine at 176.04 ppm was used as standard sample for chemical shift. The spectra were analysed by MNova 14.1.0 (Mestrelab Research).

### 4.4. Micromorphological Assessment

After freeze drying, scaffolds were removed water content by sequential ethanol series (10%, 30%, 50%, 70%, 80%, 90%, 95%, and 100% ethanol) and evaporated using an automated critical point dryer (Leica EM CPD300). Dried scaffold was fractured into small pieces and sputter-coated with gold. A JEOL JSM-6610LV scanning electron microscope was used to visualize surface feature at 1000× magnification.

### 4.5. Mechanical Assessment

The mechanical properties of scaffolds were measured by the unconfined stress-relaxation test to obtain the equilibrium Young’s modulus (E_Y_). Scaffolds (n = 4) were soaked in PBS overnight and placed in a testing chamber containing PBS at 37 °C under tare load of 50 N for 30 min. The stress-relaxation test at a ramp velocity of 2 μm·s^−1^ was performed up to 50% strain using the Universal Testing Machine EZ-S (Shimadzu, Japan).

### 4.6. Swelling Ratio

The dry weight (W_d_) of scaffolds (n = 4) was recorded, and the scaffolds were soaked in PBS solution (pH 7.4) at 37 °C for 24 h. The scaffolds were removed from PBS solution and excess liquid was blotted. The wet weight (W_w_) was recorded to calculate the swelling ratio, S, as described in Equation (1).
S = (W_w_ − W_d_)/W_d_(1)

Change in swelling ratio per unit of time, the swelling rate (S_R_), was determined when swelling ratio linearly increased, as described in Equation (2), where S_t+__∆t_ represents the swelling ratio at time t + ∆t and S_t_ represents swelling ratio at any time t.
S_R_ = (S_t+__∆t_ − S_t_)/∆t(2)

### 4.7. Cell Culture and Cell Seeding

Chondrocytes (Lonza, Walkkersville, MD, USA) were expanded in growth medium (DMEM supplemented with 1× serum replacement, 10 mM HEPES, 1% antibiotic-antimycotic solution) and maintained in a 5% CO_2_ incubator at 37 °C, with medium changes every 3 days. Subculturing was performed using TrypLE^TM^ Express Enzyme when cells reached 80% confluency. Scaffolds 8 mm in diameter and 5 mm in thickness were placed in 15 mL conical tubes. Twenty μL of 2 × 10^5^ cells was seeded on the scaffold. Cell-seeded constructs were kept in a 5% CO_2_ incubator at 37 °C for 3 h to allow cell absorption into the scaffolds. Then, 1 mL of additional growth medium was added. The constructs were returned to the incubator and continued to grow for 7 days with medium changes every 3 days.

### 4.8. Assessment of Cell Distribution and Cytotoxicity of PVA/CMC Scaffolds

At day 7 post-seeding, constructs were cut through the center and incubated in Invitrogen^™^ LIVE/DEAD^®^ viability/cytotoxicity assay solution for 30 min in the 5% CO_2_ incubator at 37 °C. The concentration of the assay solution was as described in manufacturer’s instructions: 2 mM calcien-AM and 4 mM ethidium homodimer-1. The assay solution was replaced by 2 mL of 1× PBS to remove excess dye for 30 min. The constructs were visualized under a fluorescent microscope to determine living and dead cells at wavelengths of 485 and 530 nm, respectively.

The cell viability assay was performed in a 96-well plate (cat. 167008, Thermo Fisher Scientific, MA, USA). Chondrocytes were seeded at 15,000 cells/well and cultured in 100 μL growth medium for 24 h to allow cell attachment. After that, the medium was replaced with 100 μL of the following solutions: (1) DMEM, (2) DMEM incubated with P5C1 scaffold extract, (3) DMEM incubated with P3C1 scaffold extract, and (4) 10% DMSO in DMEM. The plate was incubated at 37 °C, 5% CO_2_ for 24 h. Cell viability assay was performed using PrestoBlue™ assay (Invitrogen, Carisbad, CA) following the manufacturer’s instructions. The plate was kept at 37 °C in 5% CO_2_ conditions for 40 min. Absorbance was measured at λ_ex_ 560 nm and λ_em_ 590 nm. Scaffold extracts were prepared according to the ISO 10993-5:2009 test for in vitro cytotoxicity. Scaffolds (n = 5) were incubated in 1 mL serum-free DMEM at 37 °C, 5% CO_2_, for 24 h.

### 4.9. Statistical Analysis

Statistical analysis was performed using GraphPad Prism software (La Jolla, CA, USA). Data were expressed as average ± standard error of n = 4–6 per group. The differences in Young’s modulus, swelling ratio, and cytotoxicity were evaluated using two-way ANOVA, followed by Tukey’s post-test with α = 0.05 to consider statistical significance.

## Figures and Tables

**Figure 1 molecules-26-00578-f001:**
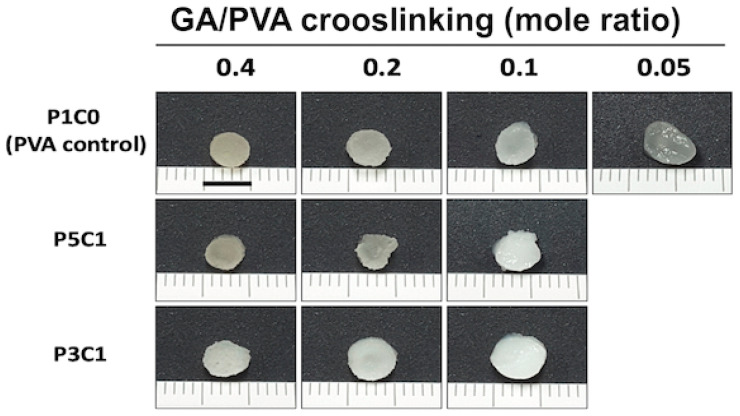
Appearance of PVA/CMC porous scaffolds. P1C0 is PVA/CMC = 1:0; P5C1 is PVA/CMC = 5:1; P3C1 is PVA/CMC = 3:1. The scale bar is 0.5 cm.

**Figure 2 molecules-26-00578-f002:**
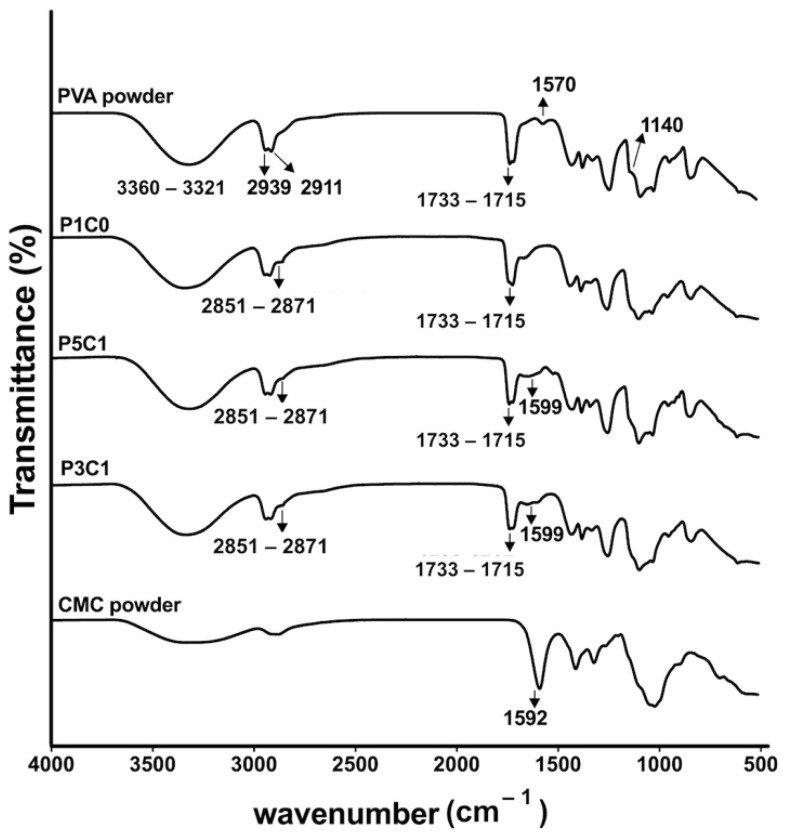
ATR-FTIR analysis of PVA powder, P1C0 PVA control, P5C1 scaffold, P3C1 scaffold, and CMC powder. The representative scaffolds have a GA/PVA crosslinking ratio of 0.2. P1C0 is PVA/CMC = 1:0; P5C1 is PVA/CMC = 5:1; P3C1 is PVA/CMC = 3:1.

**Figure 3 molecules-26-00578-f003:**
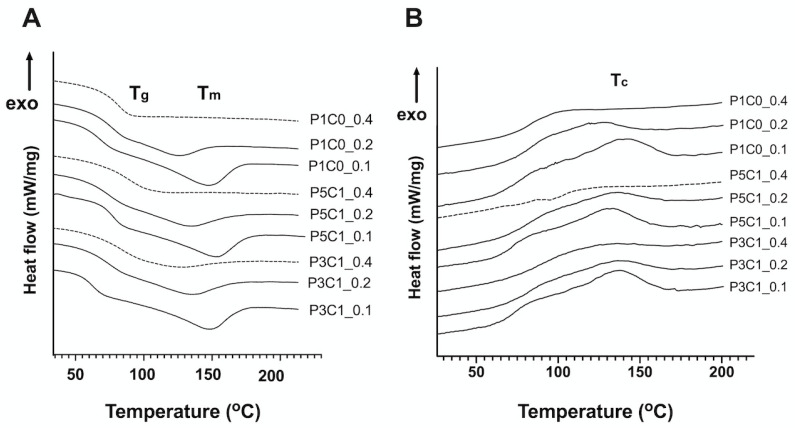
Differential scanning calorimetry (DSC) thermal analysis of scaffolds (exo up). P1C0 control, P5C1, and P3C1 scaffolds with different GA/PVA crosslinking ratios of 0.4, 0.2, and 0.1. (**A**) Heating curves and (**B**) cooling curves; dashed lines in the heating curves indicate three groups without T_m_ peaks (GA/PVA crosslinking ratio of 0.4). The dashed line in cooling curve indicates the group without T_c_ peak (P5C1_0.4). P1C0 is PVA/CMC = 1:0; P5C1 is PVA/CMC = 5:1; P3C1 is PVA/CMC = 3:1.

**Figure 4 molecules-26-00578-f004:**
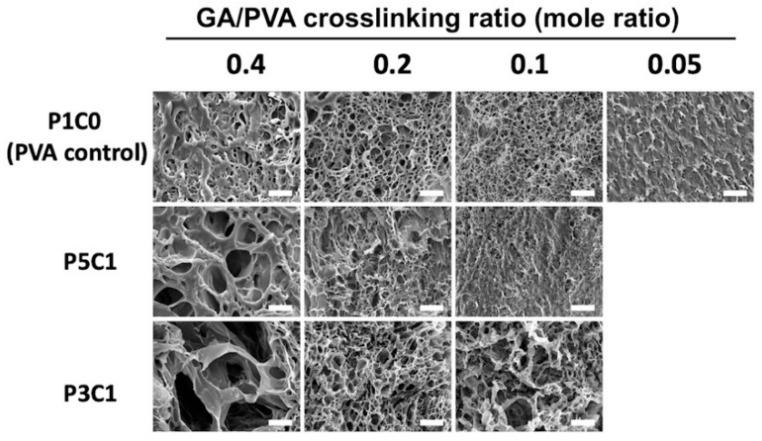
SEM images of PVA/CMC scaffolds. Hydrogels were crosslinked by GA at different mole ratios of GA/PVA and freeze-dried. PVA without CMC (P1C0) served as a control group. Scale bar = 50 µm. P1C0 is PVA/CMC = 1:0; P5C1 is PVA/CMC = 5:1; P3C1 is PVA/CMC = 3:1.

**Figure 5 molecules-26-00578-f005:**
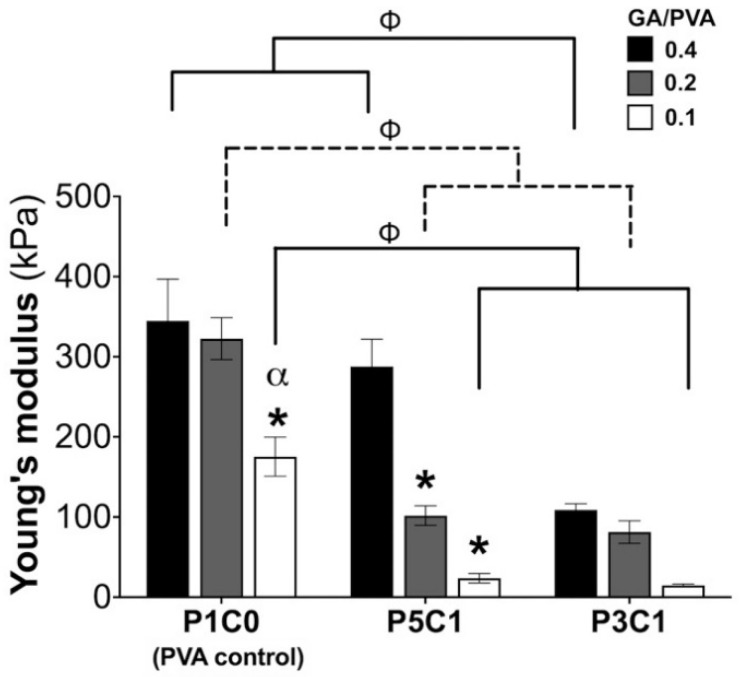
Young’s modulus of PVA/CMC scaffolds. The Young’s modulus of scaffolds of different CMC contents (P1C0 is PVA/CMC = 1:0; P5C1 is PVA/CMC = 5:1; P3C1 is PVA/CMC = 3:1) and three crosslinking ratios (GA/PVA crosslink = 0.4, 0.2, and 0.1) was determined. Data show average ± standard error; n = 4. ^ϕ^ indicates significant effects of CMC. * and ^α^ indicate significant effects of crosslinking ratios within the same group, where * and ^α^ are compared with crosslinking ratios of 0.4 and 0.2, respectively.

**Figure 6 molecules-26-00578-f006:**
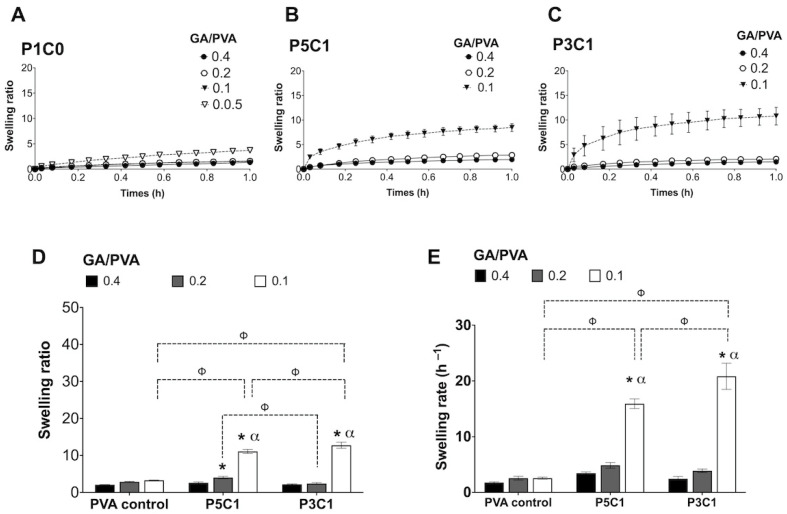
Swelling ratio. Swelling ratios of P1C0 (**A**), P5C1 (**B**) and P3C1 (**C**) were recorded for 1 h. The maximum swelling ratio at 24 h (**D**). Swelling rate of scaffolds was determined linearly over 30 min intervals (**E**). Data show average ± standard error; n = 4. ^ϕ^ indicates significant effects of CMC. * and ^α^ indicate significant effects of crosslinking compared with GA/PVA crosslinking ratios of 0.4 and 0.2, respectively.

**Figure 7 molecules-26-00578-f007:**
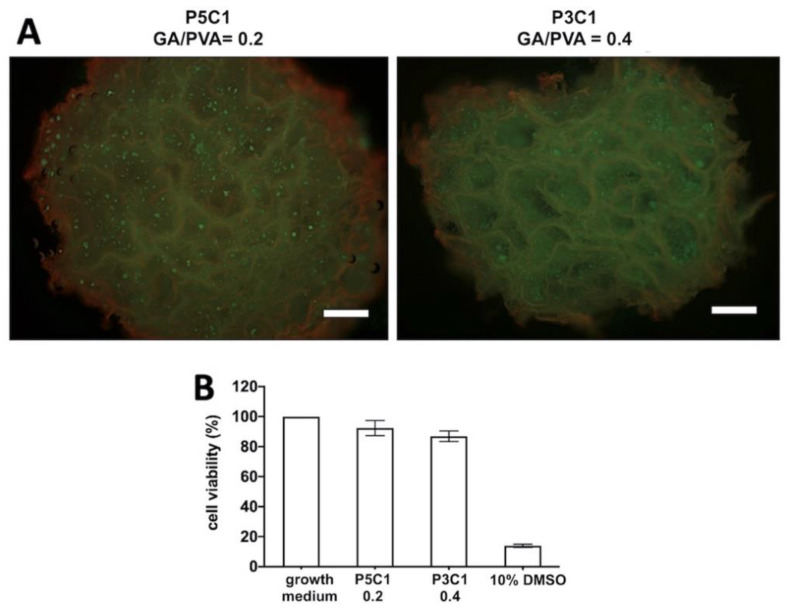
Assessment of cell distribution and cell viability. Chondrocytes were homogenously distributed in the P5C1 scaffold at a crosslinking ratio of 0.2 and in the P3C1 scaffold at a crosslinking ratio of 0.4; scale bar = 50 μm (**A**). Cell viability of chondrocytes cultured in PVA/CMC scaffold extracts (**B**).

**Table 1 molecules-26-00578-t001:** Preparation of PVA/CMC scaffolds with different glutaraldehyde crosslinking. PVA and CMC were mixed and crosslinked by GA. P1C0 (PVA/CMC = 1:0) served as control group by the addition of water instead of CMC.

Scaffolds	PVA/CMC Mass Ratio	Crosslink Ratio (GA/PVA Mole Ratio)			
		0.4	0.2	0.1	0.05
P1C0 (control)	1:0	O	O	O	O
P5C1	5:1	O	O	O	X
P3C1	3:1	O	O	O	X

PVA = polyvinyl alcohol, CMC = carboxymethylcellulose, GA = glutaraldehyde, O = gel, X = not gel.

## Data Availability

The data presented in this study are available on request from the corresponding author.

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
