# Peer review of "Carboxymethyl Cellulose Entrapped in a Poly(vinyl) Alcohol Network: Plant-Based Scaffolds for Cartilage Tissue Engineering"

_molecules, 2021, doi:10.3390/molecules26030578_

Round 1

Reviewer 1 Report

The manuscript entitled: “Carboxymethyl Cellulose Entrapped in a Poly(vinyl) Alcohol Network: Plant-based Scaffolds for Cartilage Tissue Engineering ” is a very interesting and complete study about the preparation, physicochemical characterization and in vitro cytotoxicity studies of CMC/PVA scaffolds for cartilage tissue enguneering . I recommend the publication of the manuscript after elucidate some details:

- In the physicochemical characterization of the CMC/PVA scaffolds I recommend the authors to complete this characterization with H-NMR spectroscopy, in order to elucidate the functional groups of each molecule involved in the nanosystem formulation;

Author Response

Response to Reviewer 1 Comments

We thank Reviewer 1 for constructive comments. Your comments are very important for improving our manuscript. We carefully revised the manuscript and amended the text as necessary to address the Reviewer’s points. Our itemized responses are listed below. For easy tracking, all changes in the manuscript are shown in red font.

Reviewer 2 Report

The following study investigated the combination of polyvinyl alcohol and incorporation of carboxymethyl cellulose to fabricate scaffolds potentially for cartilage tissue engineering.

Overall the paper focuses on fabrication aspects of the composite scaffold.

The cell biology associated with the study is very limited towards an indication that the material is an ideal candidate for cartilage tissue engineering.

I have detailed a large number of concerns and changes.  The written English in certain sections is limited and would benefit by an independent reader 

Review of the manuscript has the following items to address;

Abstract

Lines 17-19;

  • The following statement ‘The results suggest that PVA/CMC……..to support cartilage formation”

The study has indicated that cells seeded have remained viable for the duration of the culture, however the functional aspects of cartilage formation in particular matrix deposition have not been investigated that would support this possibility of cartilage formation.  In addition, Chondrocytes can also dedifferentiate into fibroblasts.  Cellular phenotype would also add credit to maintaining chondrocyte cells.

This final statement should be revised to include the following ‘potentially’ support cartilage formation.

  1. Introducton

Line 68

Statement that ‘Pore architecture ………..was quantified’.  This was not performed in this study and should be removed and more accurate description of what was actually performed qualitatively.

Results

Figure 4

Line 135 In Legend

  • What does this mean ‘(edit figure GA/PVA)’

Figure S4 Swelling ratios

  • No detail of Bars used in graphs, missing key

Figure S5 water content of scaffolds

  • Additional table label needed explaining what the 0.4, 0.2, 0.1 and 0 indicates

Figure S6

  • Osteosarcoma cells described in the description, not relevant to study, are these the correct data?

Figure S7

  • Missing information on glutaraldehyde crosslinking level for both samples P1C0 and P3C1

2.7. Assessment of Cell Distribution and Cytotoxicity of PVA/CMC Scaffolds

Line 185

  • Unclear on how the method was performed to assess viability/cytoxicity. If this refers to the extracts from the scaffolds, this was stated to be performed in no serum. The culture with Chondrocytes was performed in growth medium (now with serum).  This is not clear in the methods if the extract was diluted in the growth medium, which would result in not testing directly the neat extract from the 24 hour extraction step.

Discussion

Line 198

The study presented in this paper are primarily focussed on material fabrication with limited cell biology associated with cartilage formation.  The limited cell study indicates that cells can be incorporated viably but there is no data on cell proliferation or on matrix production (proteins ( collagens and Glucosamino glycans ) expected of cartilage formation)

Therefore  line 198

Scaffolds for potential use in cartilage tissue engineering…….

Line 202, 203

  • ‘We demonstrated how to control mechanical properties, swelling behaviors, and cartilage tissue development via….’

This is incorrect as there was no demonstration of cartilage development in this study

Line 207

  • What is the purpose of referencing [19]? A revised sentence structure or further explanation is needed here.

Line 215

‘Hydroxyl groups lead’ change ‘to hydroxyl groups led’

Line 217

  • What is the reason to include the statement below.

‘Besides GA, other crosslinking agents, such as epichlorohydrin [26] and citric acid [27], were used to crosslink PVA and CMC mixture under alkaline or acidic conditions at high  temperature. These crosslinking reactions resulted in different products: crosslinked PVA–PVA, PVA–CMC, and CMC–CMC.’

It needs further clarification or reference back to this reported study

Line 256

  • What is the relevance to reference [8]

Lines 284-296

  • Discussion here is rather confusing to understand. There is reference to other investigations carried out by others and then some discussion back to this reported study as of line 293.

‘Cell–scaffold interaction in this study was minimally influenced by serum protein absorption because serum-free growth medium was used during 7-day post-cell seeding’.

  • Is this statement relevant to this study or that of Ref [33]?

The step of ‘serum-free growth medium was used during 7-day post-cell seeding’ is not consistent with the methods reported in this study.

  • Please clarify methods and ensure discussion is clear

Lines 358-360.

  • This sentence needs attention.

The use of ‘…….., and it was inactivated by glycine’ needs rephrasing as it is structurally incorrect.

Line 256

What is the purpose for reference [8]

Materials and Methods

  • DSC missing method
  • NMR missing method
  • Water content experiment method is missing

In reference to Figure S6

method for Cell adhesion assay on 96 well plates is not written clearly.

  • What sort of 96 well plates were used, brand, Tissue culture plates?
  • The method describes the use of osteosarcoma cells. This is not consistent with the cells described in this study.  This cast doubt over presented data.
  • Volumes have been incorrectly described in ml units, should be in µl
  • No mention of glutaraldehyde crosslinking and if it was quenched

Line 390

  • what volume of DI was added for the soaking for 48 hours? Clarification on the freezing step, was this after the soaking for 48 hours?
  • Clarify why the UV step was used, for sterilisation? So subsequent tissue culture could be performed?

Line 399

  • What is meant by clean and dry container? How Is sterility maintained?

line 407

  • More detail is needed to describe what is meant by ‘Scaffolds were dehydrated….’ Is this after freeze drying ?

Line 410

  • ‘surface feathers” ? or ‘surface features’

Cell culture and cell seeding

  • Line 427 What is meant by 1x serum supplement.
  • What was the source of the media components and other reagents .

Assessment of cell distribution and cytoxicity of scaffolds

Line 442 and 443

  • 100 ml should be 100 µl

Line 443

Explain what is meant by (2) DMEM incubated with P5C1 scaffold extract.  Is this a dilution of the original extract set up according to ISO 10993-5 2009?  Very confusing as not clear if this is a neat extract or if it was further diluted for testing in DMEM.

In Figure 7 (B)

Lines 193 194

  • In the legend it states that the cell viability of chondrocytes was assessed after exposure to growth medium incubated with PVA/CMC scaffolds.  This is contrary to line 449 where it states that scaffolds were incubated in 1ml serum free DMEM.   This needs clarification.

Author Response

Response to Reviewer 2 Comments

We thank Reviewer 2 for constructive comments. Your comments are very important for improving our manuscript. We carefully revised the manuscript and amended the text as necessary to address the Reviewer’s points. Our itemized responses are listed below. For easy tracking, all changes in the manuscript are shown in red font.

Please see the attachment. We have used MDPI English Editing services to help us improve English language and style.

Reviewer 3 Report

Author information and affiliation are missing in the  main manuscript. It was only included in the supplementary material;

Figures 3A and 3 B should have the same size;

Figures 6A, 6B and 6C should be enlarged in comparison to 6D and 6E;

The inclusion of a graphical abstract would be appreciated;

Author Response

Response to Reviewer 3 Comments

We thank Reviewer 3 for constructive comments. Your comments are very important for improving our manuscript. We carefully revised the manuscript and amended the text as necessary to address the Reviewer’s points. Our itemized responses are listed below. For easy tracking, all changes in the manuscript are shown in red font.
